# Spatially visualized single-cell pathology of highly multiplexed protein profiles in health and disease

Mayar Allam[1,5], Thomas Hu[1,2,5], Shuangyi Cai[1], Krishnan Laxminarayanan [3], Robert B. Hughley[3] & Ahmet F. Coskun [1,3,4✉]

Deep molecular profiling of biological tissues is an indicator of health and disease. We used imaging mass cytometry (IMC) to acquire spatially resolved 20-plex protein data in tissue sections from normal and chronic tonsillitis cases. We present SpatialViz, a suite of algorithms to explore spatial relationships in multiplexed tissue images by visualizing and quantifying single-cell granularity and anatomical complexity in diverse multiplexed tissue imaging data. Single-cell and spatial maps confirmed that CD68+ cells were correlated with the enhanced Granzyme B expression and CD3+ cells exhibited enrichment of CD4+ phenotype in chronic tonsillitis. SpatialViz revealed morphological distributions of cellular organizations in distinct anatomical areas, spatially resolved single-cell associations across anatomical categories, and distance maps between the markers. Spatial topographic maps showed the unique organization of different tissue layers. The spatial reference framework generated network-based comparisons of multiplex data from healthy and diseased tonsils. SpatialViz is broadly applicable to multiplexed tissue biology.

[1] Wallace H. Coulter Department of Biomedical Engineering, Georgia Institute of Technology and Emory University, Atlanta, GA, USA. [2] School of Electrical and Computer Engineering, Georgia Institute of Technology, Atlanta, GA, USA. [3] Parker H. Petit Institute for Bioengineering and Bioscience, Georgia Institute of Technology, Atlanta, GA, USA. [4] Interdisciplinary Bioengineering Graduate Program, Georgia Institute of Technology, Atlanta, GA, USA. [5] These authors contributed equally: Mayar Allam, Thomas Hu. ✉email: ahmet.coskun@bme.gatech.edu

Single-cell, spatially resolved, multiplexed data provides the interrelations of individual marker expressions and their positional interactions for deciphering disease physiology. However, the data analytics and visualization in multiplexed data face challenges due to the big data size and biological complexity in a high-dimensional regulation using the four-dimensional (4D) information across $x$–$y$–$z$ coordinates temporal-axis. Several computational platforms are emerging, such as InsituNet[1,2], HMRF[3], Giotto[4], Trendsceek[5], and SPARK[6], to provide multiplexed imaging data analysis tools. These methods compute the statistics, abundance, relationships among multiple markers, visualize marker correlations and associations as networks, graphical plots, and statistical representations. InsituNet visualizes the coexpression between individual transcripts by converting them into interactive spatial networks. HMRF identifies cellular subpopulations and overlays cluster information with spatially preserved maps for single-cell visualization. Giotto is a broadly applicable tool that enables spatial data visualization from several multiplexing modalities, including fluorescence in situ hybridization (FISH), proteomic multiplex imaging, and sequencing[4]. SPARK identifies spatially resolved transcriptomics profiles and provides statistical analyses for the spatial correlations[6]. Data reduction analyses using principal component analysis (PCA), t-distributed stochastic neighbor embedding (t-SNE), and Uniform Manifold Approximation and Projection (UMAP) methods yield cellular phenotypes in the multiplexed datasets[7,8]. While toolsets are being developed, spatial visualization of protein datasets necessitates a cross-scale spatial hierarchical analysis to link the single-cells to tissues' anatomy.

Deciphering tonsil biology using multiplexed proteomic imaging and emerging data analysis is crucial. Tonsils are part of mucosa-associated lymphoid tissue, playing a vital role in the immune system, and generally act as the first-line defense barrier to inhaled or ingested pathogens. T- and B-cells are the most predominant cell type in tonsil tissues in coordination with other immune and epithelial cells. They are primarily located around the germinal area of the tonsils. B cells help recognize the foreign antigens through the secreted antibodies and continue to increase in their density. Among other tonsil origins, palatine tonsils are located at the rear of the throat (pharynx) as a pair of soft tissue masses on both ends of the mouth studied in this study[9]. Given the immune-rich environment of tonsils, tonsil tissue analysis can be used to study ample diseases such as digestive tract infections[10], autoimmune diseases[11], leukemias[12], and respiratory diseases[13]. Thus, tonsil tissues hold a wealth of information about individuals' immune profiles and can guide the diagnosis of distinct disorders to design an appropriate treatment regimen.

Multiple immune markers need to be detected from the same tissue samples to unleash the inflammatory information presented in the tonsil. Traditionally, clinicians relied on histological images that could only visualize a few markers at a time. Multiplex imaging modalities overcome this marker limitation. For instance, imaging mass cytometry (IMC) is an emerging technology that relies on mass spectrometry and time of flight (TOF) measurements, wherein antibodies against antigens of interest are conjugated with isotopes of pure metals. The stained samples are then ablated by an ultra-violet (UV) laser beam, which results in aerosol plumes. These plumes later get ionized through plasma and get directed through argon and helium gas flow to the mass spectrometer, where the metal tags get analyzed based on their mass-to-charge ratio and abundance[14]. Although this multiplexed power may be partially obtained by fluorescence cytometry, the fluorophores used to detect biomarkers of interest have overlapping spectra leading to signal spillover between target and non-target detectors. That necessitates the use of additional signal processing techniques to correct for fluorophores signals in non-

target detectors, often referred to as spillover compensation. Signal compensation can be more complicated and often lead to erroneous results[15,16]. Besides, multiplexed cell suspension cytometry techniques suffer from their inability to analyze biomarkers in situ from patients' samples, resulting in the loss of crucial spatial single-cell data. Several fluorescence-based multiplexed techniques were later developed and tested for patients' archived FFPE samples, including cyclic immunofluorescence[17]. These methods rely on antibody staining, bleaching, and imaging cycles that suffer from sample autofluorescence, epitope loss, and laborious procedures. On the other hand, IMC detects the abundance of metal isotopes conjugated to antibodies against multiple biomarkers. These lanthanide isotopes are rare elements that would not be found in biological samples, resulting in high signal specificity[18]. The signal is further amplified by polymeric metal-chelating reagents, allowing a range of proteins to be detected. Preserving in situ cell–cell interaction and tumor microenvironments, IMC relates the metal isotope abundance to their pixel location, providing images of multiple proteins simultaneously in archived patients samples at subcellular resolution (1-μm)[18–20].

IMC is widely used in many applications, including drug testing and tumor studies. IMC was previously used to investigate the potential benefit of the chemotherapeutic agents on three different cancer types. A multiplex antibody panel was designed to target several proteins associated with common cancer oncoproteins and additional markers to detect T-cell infiltration (CD8) and the epithelial organization (Beta-catenin and Pancytokeratin)[21]. Further, IMC was used to analyze more than 15 proteins associated with different stages of multiple sclerosis (MS), identifying a unique subset of T-cell phenotypes associated with different MS stages[22]. Besides, IMC was used to decipher the breast cancer tumors using a panel of 35 biomarkers associated with breast cancer subtypes, grades, signaling, oncogenes, and epigenetics. Remarkably, different breast cancer subtypes with unique cellular phenotypes were strongly correlated with survival[19]. IMC revealed mRNA-to-protein associations at the subcellular level in breast cancer patient samples[23]. RNA probes and their encoding protein antibodies were conjugated to different metal isotopes to study their correlation at the population level. This growing body of discoveries highlights the importance of IMC technology to reveal interdependencies among cell types and provide spatial maps for multiple subcellular resolution parameters.

Motivated by these studies, we used IMC to profile the tonsil tissues using a 20-plex antibody panel designed to detect immune cells, epithelial cells, extracellular matrix, and functional signatures. This antibody panel was used to stain two types of tonsil samples that include three tissue sections from a disease-free subject and three tissue sections from a patient diagnosed with chronic tonsillitis. The resulting data yielded the coexpression profiles of immune markers in health and disease. These datasets provided testbeds for the development of spatial visualization tools.

Here we present SpatialViz, a series of computational analysis methods that combine two data analytics methods to bridge the gap between the top-down anatomical architecture and the bottom-up cellular assembly. The bottom-up approach relied on the segmentation of single-cells, followed by phenotypic clustering and correlation analysis. This pipeline highlighted the biological significance of correlations among the markers in the multiplexed protein panel. The bottom-up approach was further supplemented by the top-down approach used to cluster the multiplex images at the pixel level and define their corresponding anatomical structure. This approach provided single-cell enrichment maps for each marker at distinct anatomical regions along

with their morphological distributions. Unique sets of markers expressed in a subpopulation of cells were spatially associated with the network analysis of anatomical clusters. Spatial proximity maps were used to identify the distance as a metric between the cells expressing specific pairs of markers. Spatial topography maps were used to visualize the layering of marker enrichments across tonsil tissues. The spatial reference approach was then used to compare the multiplexed protein datasets from healthy and diseased samples on a single network map using landmark and moving nodes with expression levels and inter-marker distance statistics. Spatial visualization of multiplexed protein images provides a general framework that can be applied to quantify and monitor cellular changes for a wide range of tissues in health and disease.

## Results

**Single-cell quantification of immune-epithelial landscape alterations.** The acquired IMC data was first analyzed by the bottom-up approach, where a standard data analysis pipeline was used in Cell Profiler[24] and HistoCAT[25] (Fig. 1a, b). First, the raw data in the MathCad document (MCD) format was processed using the MCD Viewer to extract the data from all regions of interest (ROIs) as a separate MCD file, followed by exporting the data to individual TIFF images for each marker. These marker images were then segmented by the CellProfiler software to identify single-cell positions and expression levels. After generating the cell segmentation mask, the original images and their segmentation masks were imported into the HistoCAT to create cell phenotype clusters, cell-to-cell correlations, and t-SNE maps.[26] For example, a tissue image from chronic tonsillitis was shown in multiple synthetic colors. The E-cadherin, high-Granzyme B (GrB) crypt region, and intercalator-high nuclear boundaries were highlighted in red, green, and blue colors, respectively (Fig. 1c). These three markers were also segmented to highlight cell boundaries for cytoplasm and nucleus (Fig. 1d). Eight out of twenty markers in the multiplexed panel were uniquely grouped on t-SNE clusters with distinct colors in healthy and diseased samples, yielding significant changes for GrB, Vimentin, and CD68 (Fig. 1e).

The resultant markers' clusters showed cell distributions' heterogeneity in normal and diseased tonsil tissues (Fig. 2a, b, and Supplementary Fig. 1a, b). DT1 showed 21, DT2 exhibited 22, and DT3 revealed 26 clusters based on the unsupervised clustering performed on HistoCAT. Each of these clusters corresponds to a different phenotype in the tissue section based on several markers' coexpression. The heatmap distribution summarized the clusters and the level of expression of its constituting markers. Cluster 7, 20, and 11 of DT1, DT2, and DT3 demonstrated the coexpression of GrB and CD68, respectively. Therefore, we analyzed their coexpression at the single-cell level and found out that they were highly co-expressed at several regions of the ROIs (Fig. 2d and Supplementary Fig. 2c-d). The correlation analysis showed that the majority of diseased ROIs showed a strong correlation between GrB and CD68 with R values 0.34, 0.57, and 0.64 for DT1, DT2, and DT3, respectively (Fig. 2d and Supplementary Fig. 2c-d). Furthermore, coexpression of CD3, CD4, and CD8α markers was observed in all diseased ROIs (Fig. 2f and Supplementary Fig. 3c-d).

On the other hand, the clustering in normal ROIs showed significant differences (Fig. 2a and Supplementary Fig. 1a). NT1, NT2, and NT3 showed 23, 25, and 22 different clusters from the three normal ROIs, respectively. This clustering was demonstrated on the heatmap for the distribution of all markers. GrB was co-expressed with CD68 in clusters 21, 18, and 18 for NT1, NT2, and NT3, similar to the diseased ROIs. However, the level of

CD68 expression was significantly lower than that of the diseased ROIs. Thus, we performed a similar analysis to investigate the differential correlation between CD68 and GrB in the healthy and diseased tonsil. The coexpression and correlation level were significantly lower in all NTs, as indicated by all the R values of 0.20, 0.24, and 0.19 for NT1, NT2, NT3, respectively (Fig. 2c and Supplementary Fig. 2a-b). The visual representation of the coexpression of CD3, CD4, CD8α markers in the original images of the normal ROIs (Fig. 2e and Supplementary Fig. 3a) and their correlation (Fig. 2e and Supplementary Fig. 3b) showed that most of the CD3+ cells are CD8α+ cells in contrast to the finding from the diseased tonsil tissues.

By quantifying the single-cell data, intensity levels exhibited fewer variations in normal compared to diseased tissues. Overall, the intensity analysis in the box plots showed large variations in the dataset (Supplementary Data 1). The intensity distribution demonstrated wide expression levels for all markers, which could be attributed to the heterogeneity in the tissue samples' phenotype composition. CD20, CD68, and GrB showed higher staining intensity in the diseased tonsil than the normal tonsil (Fig. 2g). Furthermore, the percentage count was defined as the total number of cells expressing a certain marker divided by the total number of cells expressing at least one marker. Pankeratin, CD68, and GrB showed a higher percentage count in the diseased condition than the normal baseline condition (Fig. 2h). The trend of the CD68 and GrB markers distribution matched our expectations because inflammation and induced apoptosis showed elevated expression levels in the case of diseased tonsils. Pankeratin was increased with the disease condition due to the accumulation of crypt regions. These biological observations could also be dependent on the tissue condition (preservation conditions and storage period), the donors' conditions (age, gender, and physical status), and the selection of ROIs.

GrB is part of the granzymes family secreted and stored in the cytotoxic granules of the cytotoxic lymphocytes. GrB is predominantly expressed by cytotoxic lymphocytes and natural killer (NK) cells. At the same time, it can also be detected in the non-immune cell types, including smooth muscle cells, keratinocytes, and chondrocytes. Other immune cells express GrB in pro-inflammatory conditions, including CD4+ cells, activated macrophages, mast cells, neutrophils, and basophils[27,28]. Monocytes target antibody-coated pathogens through antibody-dependent cell-mediated cytotoxicity (ADCC) mechanism primarily regulated by GrB expression to induce cell death[29]. Thus, the observed link between monocytes and GrB in our dataset can be used as a differentiator of health and disease. We used CD68 as a macrophage marker because it has classically been used to identify macrophages as a prognostic marker for cancer progression[27]. As previously noted, we observed a higher correlation between CD68 and GrB in the diseased ROIs than in the healthy tissue's ROIs. On a separate note, CD4+ cells are the central regulators of immune responses such that they can differentiate into specialized effector cells once they get activated by pathogen exposure. For example, CD4+ cells have a central control in autoimmune disorders such as rheumatoid arthritis and are present more predominantly than CD8α+ cells[28]. Our data support these findings as most of the CD3+ cells are also CD4+ cells in the diseased condition.

Furthermore, the Forkhead Box Transcription Factor P3 (FOXP3) and CD4 markers showed a strong correlation in the case of the normal and the diseased ROIs, indicative of regulatory T (Treg) cells (Supplementary Fig. 4). FoxP3+ cells are a subset of CD4+ cells because the generation of T cells with a suppressor function (FOXP3+ cells) occurs when dendritic cells fail to activate the CD4+ cells[30]. FOXP3 is an established marker to identify Treg cells; however, other markers are also commonly

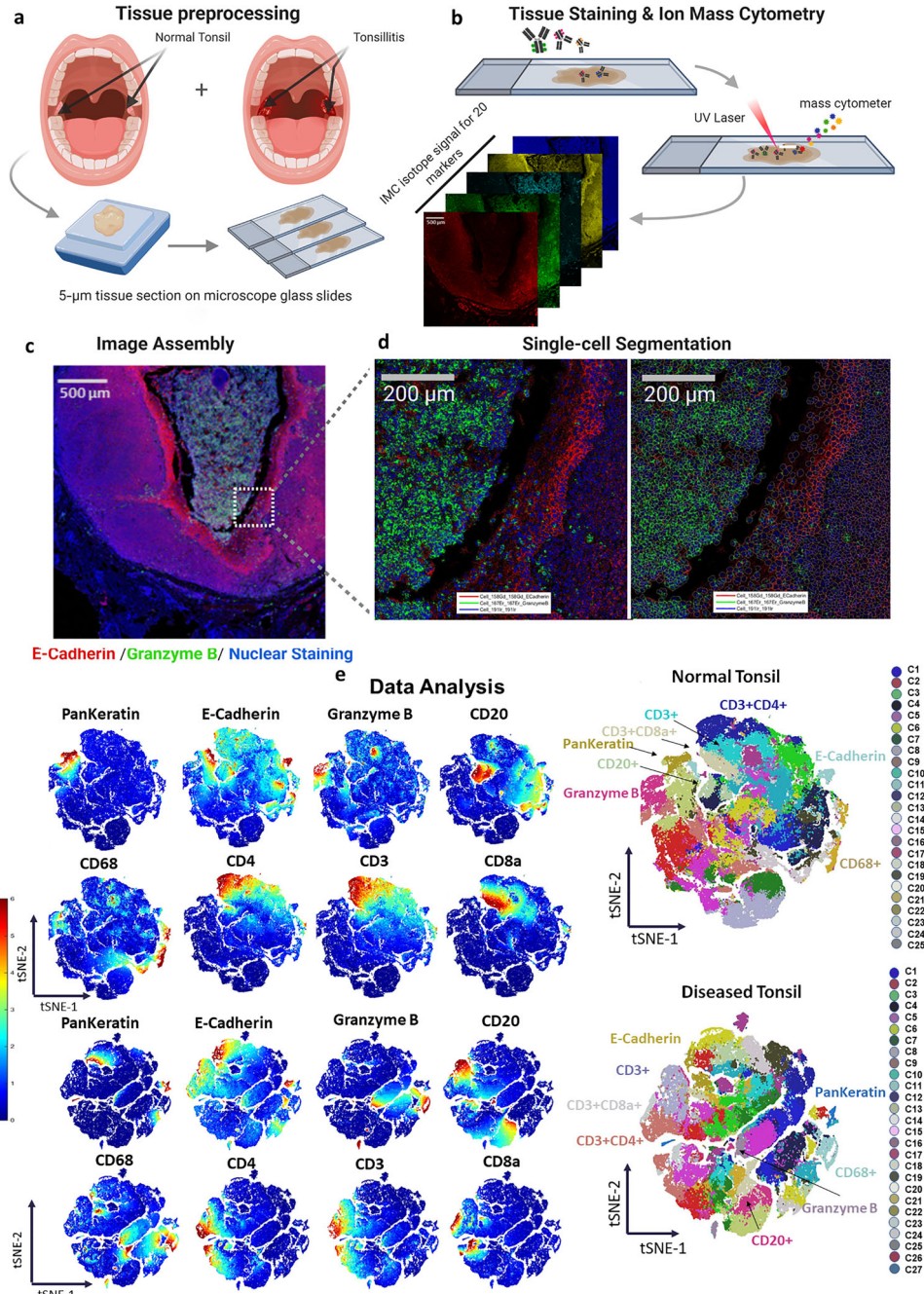

**Fig. 1 The immune-epithelial landscape of healthy tonsil and chronic tonsillitis. a** Tissue samples were obtained from healthy and diseased tonsils (chronic tonsillitis) in 5-μm-thick formalin-fixed, paraffin-embedded samples. **b** The thin tissue sections were then stained with metal-conjugated antibodies and get ablated from the tissue's surface through the argon plasma and analyzed by the time of flight mass spectrometer. **c** Individual marker images can be assembled and visualized using different colors to observe the marker's coexpression and spatial organization. **d** Cells were segmented using CellProfiler software for single-cell quantification. **e** Single-cell analyses provide the marker expression distributions, including phenotype clustering based on the coexpression of markers. Phenographs of all normal and diseased ROIs show the different phenotypes that make up the sample to $n = 25$–27 groups. Immune and stromal markers were selected to cluster the entire dataset of the normal and diseased tonsil. The markers list includes CD20, CD68, CD3, CD4, CD8a, granzyme B, pankeratin, and E-cadherin. Created in Biorender.com.

used to identify them, including CD25, CTLA-4/CD152, CD27, OX40, CD62L, CD39, and CD44[31–34]. Thus, we analyzed both FOXP3 and CD44 expressions in our dataset concerning the CD4 marker (Supplementary Fig. 5). FOXP3 was co-expressed with CD4 in several clusters both in the normal and the diseased tonsil samples (Supplementary Figs. 1a-b). We conducted a correlation analysis between both FOXP3 and CD4 and CD44 and CD4. Although both pairs (CD4 and FOXP3; CD4 and CD44) showed

a significant correlation, our dataset yielded no substantial change in the normal and diseased tonsil datasets (Supplementary Fig. 6). Using CD44 and FOXP3, Treg cells exhibited higher expression in normal conditions than the diseased conditions, agreeing well with the observation that Treg cells can suppress the immune response (Supplementary Fig. 6).

Besides, Treg cells may exhibit a memory phenotype and express CD45RO[35]. Correlations between CD44 and CD45RO

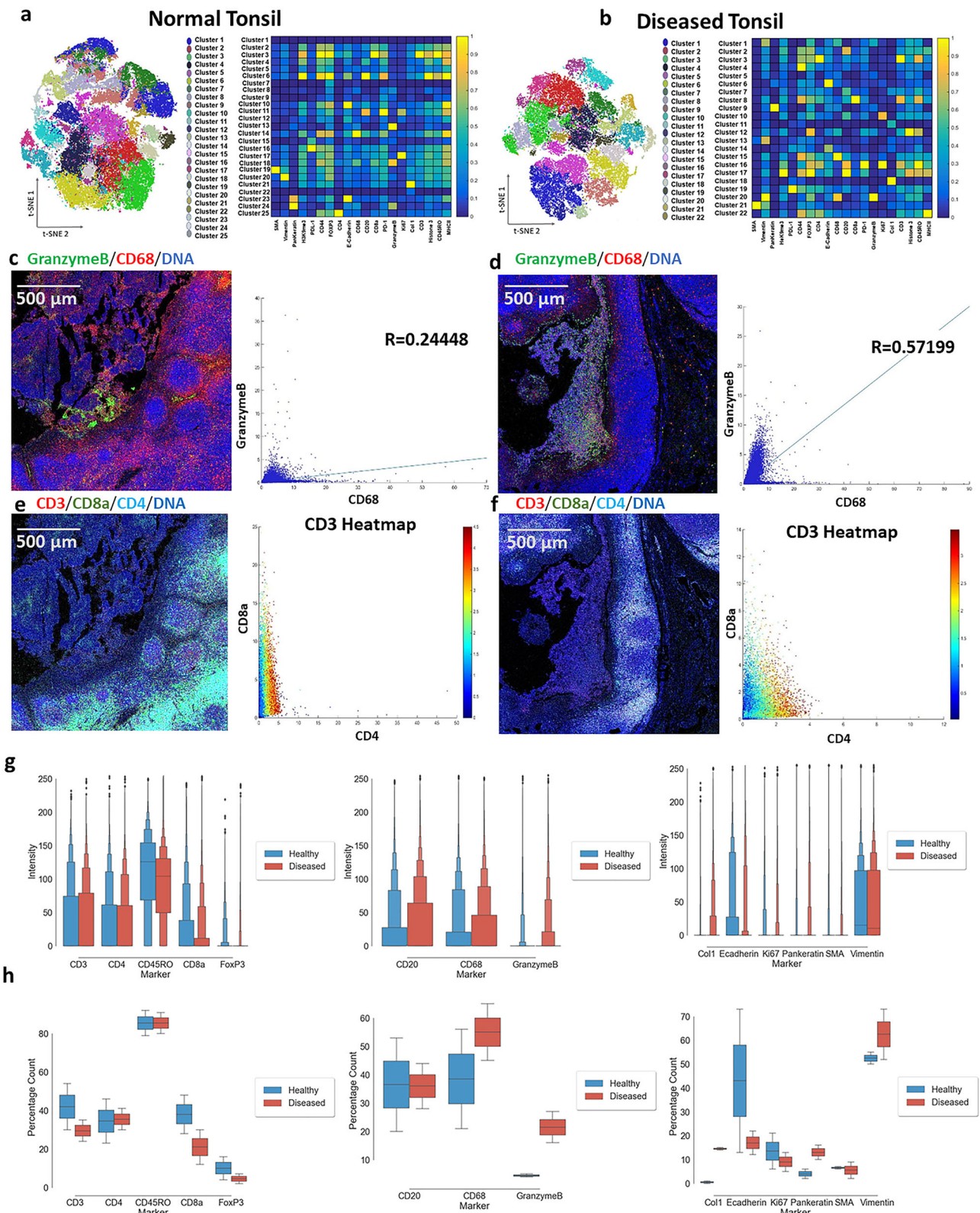

and FOXP3 and CD45RO were computed to assess Treg cells phenotype (Supplementary Fig. 7). Supporting prior findings, CD44 and CD45RO exhibited a strong correlation in the heatmaps by cluster 17 and cluster 26 (Supplementary Fig. 1b DT2 and DT3) and cluster 1 and cluster 3 (Supplementary Fig. 1a NT1 and NT2)[31]. Correlation analysis on the single-cell expression levels of CD44 and CD45RO demonstrated a strong correlation between the two markers in the diseased and the normal tonsil (Supplementary Fig. 7). Finally, the correlation analysis between CD44 and FOXP3 was weaker than the CD44 and CD45RO (Supplementary Figs. 6, 7).

PD-1 and PD-L1 expressions were weak in our data, but PD-1 showed a detectable expression at the germinal center (Supplementary Fig. 8). Prior reports observed PD-1 and PD-L1 expression in

**Fig. 2 Immune cells are altered in chronic tonsillitis when compared to healthy tonsils. a, b** Phenograph and heatmap show the different clusters' distributions in the normal and diseased tonsil datasets. **c** Visual representation and the correlation analysis between Granzyme B and CD68 in one normal tonsil ROI. The correlation between granzyme B and CD68 was found to have Pearson's correlation coefficient of $R = 0.24448$ for normal tonsil ($n = 56,421$ cells) and $R = 0.57199$ for diseased tonsil ($n = 56,421$ cells). $R$ values calculate the z-score through the Fisher Z-Transformation, resulting in a p-value $< 0.001$ (***) indicating a significant statistical difference between them. **d** Visuals and the correlation plot between Granzyme B and CD68 in one diseased tonsil ROI are shown. **e** Pictures and the correlation graph between CD4, CD8a, and CD3 in one normal tonsil ROI are presented. **f** Images and the correlation analysis between CD4, CD8a, and CD3 in one diseased tonsil ROI are demonstrated. **g** Box plot indicates the intensity distribution for all markers in the dataset for normal (blue) and diseased (red) tonsil at the single-cell level ($n = 446,123$ cells). The difference in the box plot had a significant p-value $< 0.001$ (***) by the Kolmogorov–Smirnov statistic (two-sided). The first box covers the central 50%, and the second box extends from the first to cover half of the remaining area (75% overall, 12.5% leftover on each end). The third box covers the remaining area (87.5% overall, 6.25% left on each end). This procedure is repeated until the leftover points are marked as outliers. **h** Box plot demonstrates the percentage count for all markers in the dataset for normal (blue) and diseased (red) tonsil. Box plots show median first and third quartile, minimum and maximum (excluding outliers).

the tonsil and other lymphoid tissues, including the thymus and the spleen[36,37]. For example, PD-1 was preferentially present at the tonsils' germinal center and exhibited coexpression with the CD3 marker mostly by the germinal center-associated T cells[36]. The observed weak PD-1 staining could be attributed to the PD-1 and PD-L1 antibodies' clonal specificity and the tissue type[37].

Finally, the CD20 marker for B cells was expressed around the germinal center[36]. CD20 is an established marker that reveals several B cells in the tonsil tissue that can also secrete antibodies. CD3 coexisted with CD20 in the same cluster 16 in DT2 and cluster 26 in DT3 (Supplementary Fig. 1a-b). This coexpression of CD20 and CD3 was only present in the case of diseased tonsils that may be attributed to CD3+ cells activating antigen-specific naive B-cells in the germinal centers in response to infections[38]. Finally, pankeratin was observed to be co-expressed with E-cadherin (Supplementary Fig. 1a-b). Pankeratin serves as a marker for the surface epithelial layer of the tonsil tissue[39]. In our dataset, the surface epithelial layer also shared a strong coexpression pattern of E-cadherin. These findings show the alterations of immune and epithelial landscape in the multiplex tissue images from healthy and diseased subjects. Such quantitative cellular profiling sheds light on single-cell differences using the bottom-up approach of cellular distributions.

**SpatialViz anatomy of highly multiplex tissue data.** Specific marker expression levels from different tonsil conditions exhibited heterogeneity at the single-cell level based on where they are located on tissue specimens as the anatomical features (Supplementary Fig. 9). A 20-plex multiplexed data was clustered at the pixel level by an unsupervised k-means method to generate six unique spatial patterns across the tissue that may be used as an indicator of anatomical similarity in tissues (Fig. 3a). The top-down strategy provided the spatial regions demonstrating unique morphological shapes and cellular compositions. The enrichment of a specific cell type in one of these six spatial anatomical regions is a key metric for determining structural tissue compositions. Thus, we defined an "area ratio" calculated by the area of a single marker's expression divided by the entire image area using the summation of pixel number distributions. The area ratio was between 0 and 1 displayed on a grayscale marker mask. Using this spatially resolved anatomical definition, the evidence of enrichment of both CD68 and GrB marker expressions in the diseased tissues was recapitulated, as the area ratios for CD68 and GrB were higher in datasets from diseased tonsil compared to normal tonsil (Fig. 3b, c and Supplementary Figs. 10-11). The high morphological overlapping regions also agreed well with the spatial correlation of GrB and CD68 in normal and diseased tonsils (Supplementary Fig. 12). Another cell density measure of Histone3 exhibited an area ratio comparable between healthy and disease tonsil images (Fig. 3d and e). Apart from areas, the spatial classification method included DNA1, DNA2, and Histone3

markers in one of the clusters, while CD3, CD4, and CD8α were grouped in another cluster (Supplementary Fig. 13a-b). In brief, SpatialViz groups marker images into distinct clusters based on the similarity in marked region spread, region shape, and pixel intensity. Each cluster represents a unique colored mean image calculated from all markers in that cluster and then combined in one averaged image that showed the unique anatomical traits from each cluster (Supplementary Figs. 13-14).

SpatialViz visualizes tissue anatomical representations by a network graph (Fig. 4a). To dissect single-cell distributions in each of these anatomical classes, we calculated individual cells' spatial data from the top-down analysis using segmentation data from the previous section to integrate the cellular coordinates with anatomical clusters. The relative distances among the single-cell positions in each anatomical cluster were mapped as a spatially visualized intra-clustering method. The intra-cluster distance visualizes only inside each cluster and measures the k-nearest neighbors (k-NN) distance across a ten-cell radius between each pair of markers (Fig. 4a). Besides, relative cell-to-cell separations from one anatomical group to another were referred to as inter-cluster spatial maps. Inter-cluster distance refers to the average k-NN distance of cells between two distinct clusters by taking all possible pairs of markers from the two clusters. The edge color represented the normalized distance, with red being relatively close and blue distant. The size of the node indicated the average area ratio in the cluster or for a marker. While markers exhibited significant similarity within-cluster grouping in a specific dataset, inter, and intra-cluster spatial organization differed between tonsil datasets (Figs. 4b and 5). Obvious marker grouping was consistent across datasets (DNA1, DNA2, and Histone3) with close spatial connectivity. The spatial distance network map showed high variance due to the differences in different tissue images' organization.

Of note, our anatomical spatial networks are different than those in InsituNet[1], as the current datasets are protein images compared to the in situ transcript images used in that platform. In the InsituNet pipeline, the network representation models the number of transcript detections (for node size) and the transcript's coexpression (for edge size). However, it lacks the single-cell locations and spatial data not included in the coexpression for edge representation. Thus, SpatialViz is a different approach that performs spatial anatomy network visualization of marker expression by analyzing both the single-cell level data and tissues' anatomy. The SpatialViz network model also leverages the k-NN distance between marker-specific cells and also the marker expression area.

**SpatialViz proximity map of nearest neighborhoods in tissue marker pairs.** While prior single-cell studies have focused on colocalization as a measure at the same pixel location of cell comparisons, SpatialViz computes the relative distances between

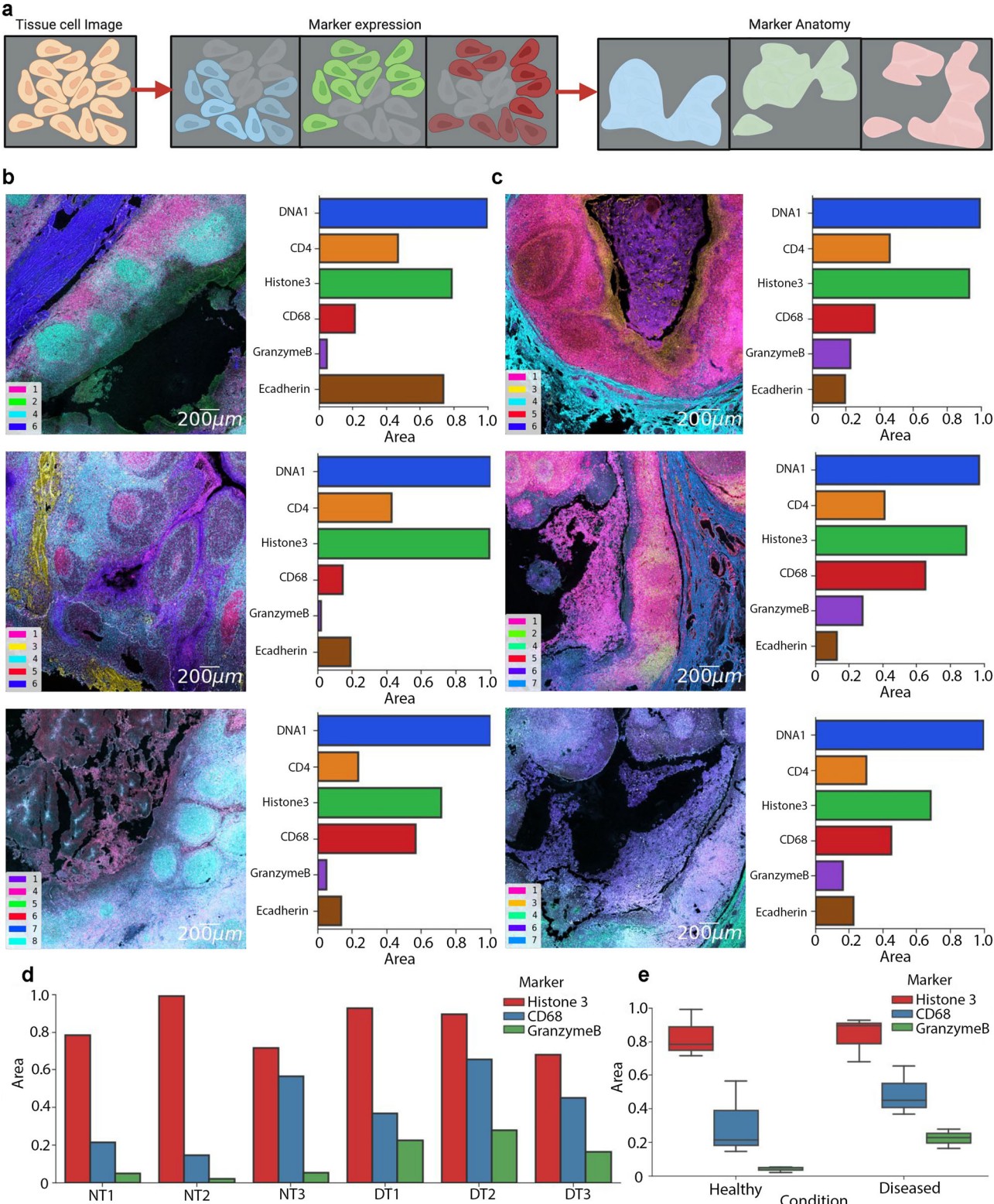

**Fig. 3 CD68 cells and Granzyme B marker coexpression and large morphological shared presence in chronic tonsillitis. a** Schematics of the definition of anatomical regions based on multiplexed marker data are presented. Created in Biorender.com. **b** Morphological analysis of area ratios of DNA1, CD4, H3, CD68, Granzyme B, and E-cadherin markers in normal tonsil tissues is demonstrated. The scale bar is 200 μm. **c** Morphological coverage of the same subset of markers in diseased tonsil tissues is shown. **d** Bar plots for area ratios of Histone3, CD68, and Granzyme B markers in three normal tonsils (NT) and three diseased tonsils (DT) datasets are plotted. **e** Box plots of the mean area ratio of Histone3, CD68, and Granzyme B markers that were averaged from three sets of NT and DT data. Box plots show median first and third quartile, minimum, and maximum (excluding outliers).

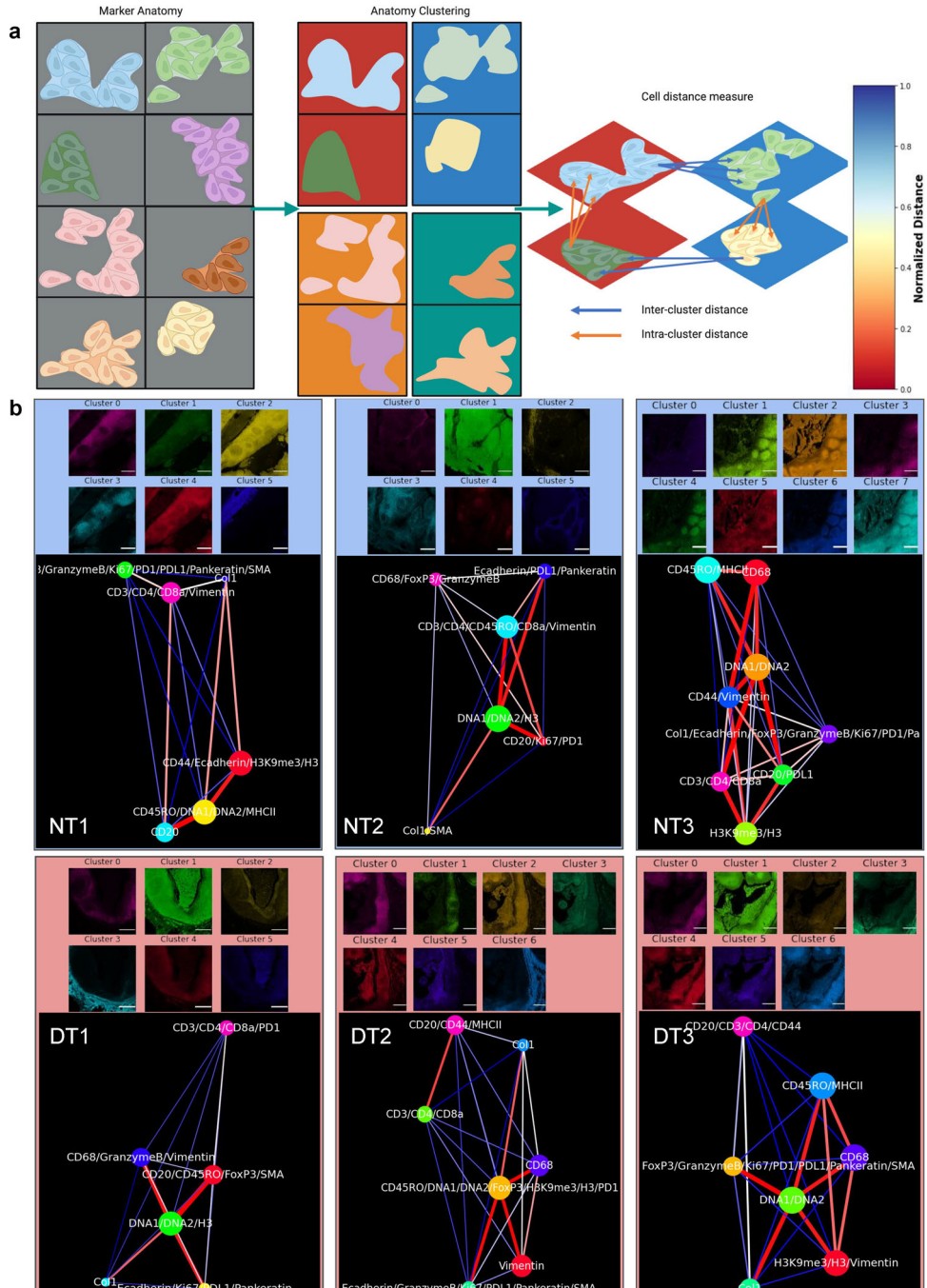

**Fig. 4 Inter-cluster distances across anatomical classes were represented as spatial network maps of single-cell positions and expression levels.**
**a** Schematic representation of intra-cluster and inter-cluster models among the single-cell locations is presented. Unsupervised clustering was performed on multiplexed IMC data to define anatomical regions. Spatial network visualization plots of marker pairs from intra-cluster (denoted with orange arrow) and inter-cluster (indicated with blue arrow) using relative distances between single-cell positions for each marker (scale bar 500 μm) are shown. Created in Biorender.com. **b** Inter-cluster distances provide spatial network maps of single-cell pairs across anatomical clusters computed for normal and disease tonsil datasets. Red edges between network nodes show short average distances, whereas blue ones demonstrate relatively distant average measurements (scale bar 500 μm).

the cells that express unique marker pairs. To visualize the pixel distances between a pair of markers, we plotted a "spatial proximity map" of the nearest neighbor cells for pairs of tissue markers and their original spatial positions (Fig. 6a, b). In this visualization, each cell from the first marker was connected to the nearest neighbor's cell from the pair's second marker. Therefore, if two markers were highly correlated spatially, then the average distance was smaller on the map. On the other hand, if two

markers were distant, the mean distance was larger in the spatial proximity analysis. The green and magenta colors showed the cells from distinct markers, and the blue line connected each cell from the first marker to its nearest neighbor cell in the second marker. The histogram on the bottom of each spatial proximity map denoted the distribution of the closest distances. A heatmap representation of these distances average provided both the spatial proximity and the density between markers (Fig. 6c and

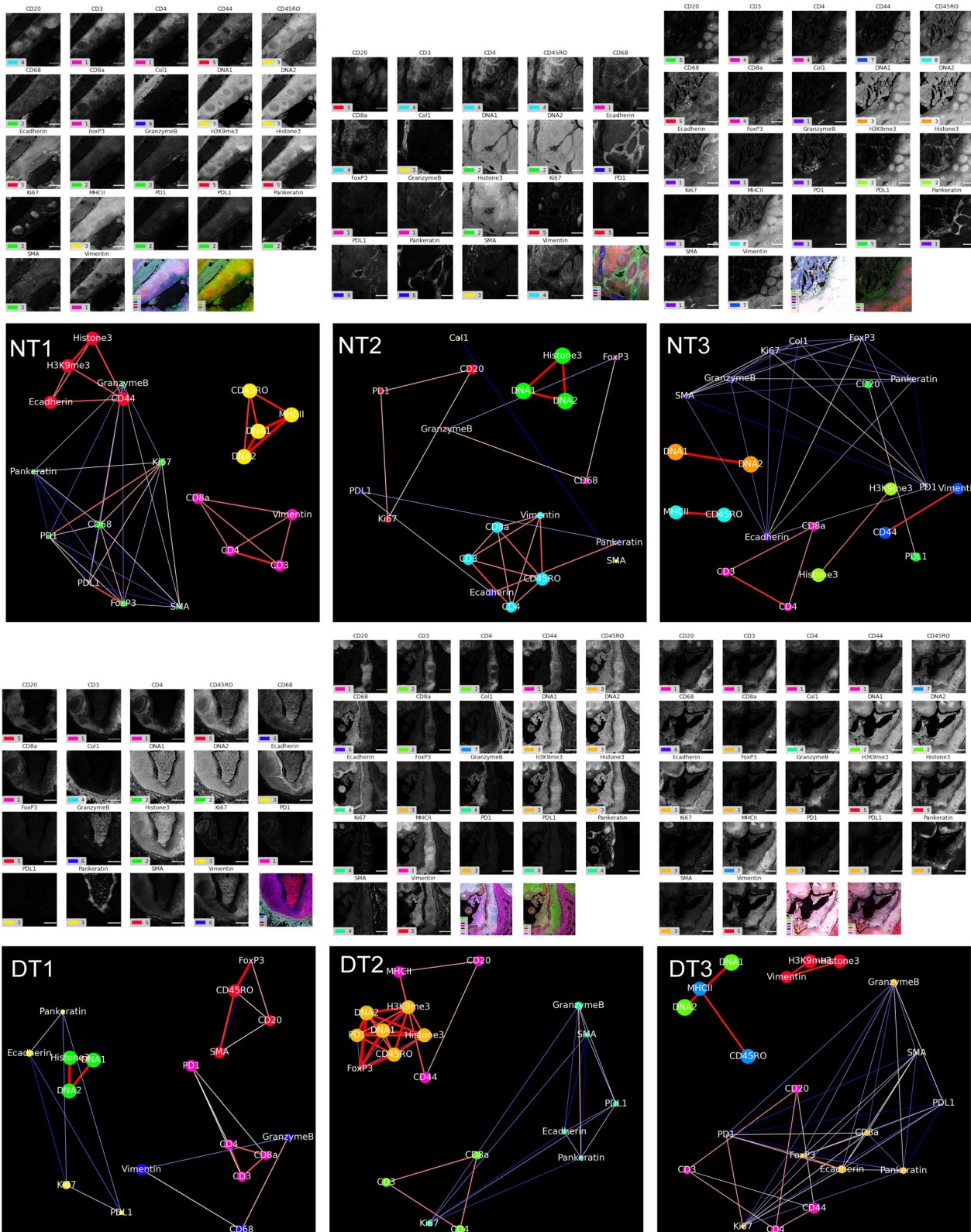

**Fig. 5 Intra-cluster distance maps of markers that are expressed in single-cells from each anatomical cluster.** Intra-cluster average spatial distance network maps are presented for single-cells from multiplexed markers inside anatomical clusters for each multiplexed image pair in healthy and diseased tonsils. The number of markers is 19–22. The color of nodes represents the corresponding cluster of the marker images. Here, only markers within the same cluster determined by anatomical clustering are linked. The red edges between network nodes show short average distances, while blue ones indicate relatively distant marker pairs (scale bar 500 μm).

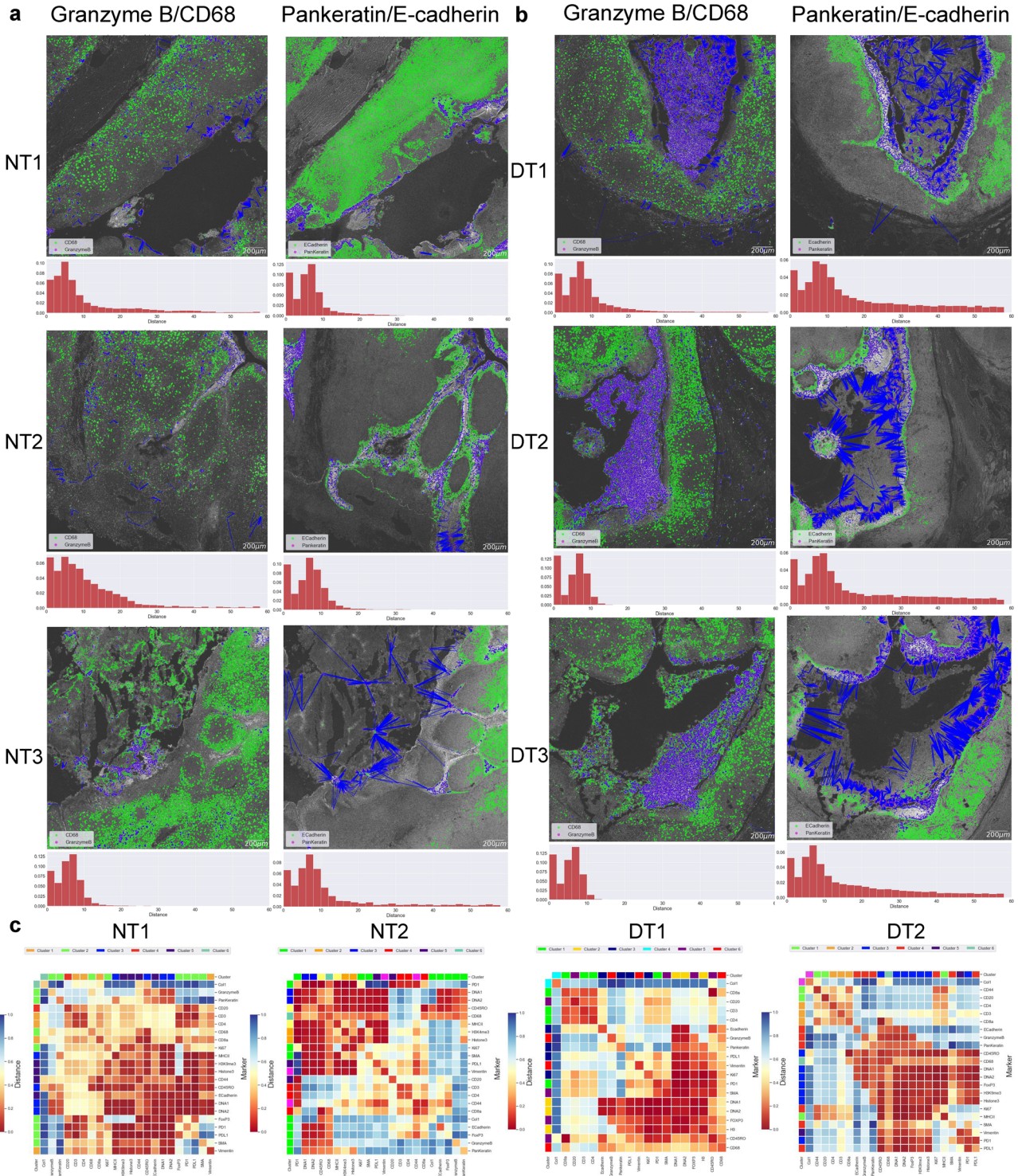

**Fig. 6 Spatial proximity maps of cell-pairs within a k-nearest neighbor (k-NN) distance in healthy and disease tonsil tonsils. a, b** Spatial proximity maps of cell-pairs within k-NN of 5-cell radius distance in healthy and diseased tonsils are visualized. Magenta and green color represent individual cells in the pair of markers (Granzyme B/CD68 and Pankeratin/E-cadherin), wherein the blue is the distance vector between the marker pairs for each cell. The scale bar is 200 μm. The histogram shows the distance distribution for each pair of markers, with the y-axis showing the proportion corresponding to the distance on the x-axis. The x-axis is distributed in 20 bins of data. **c** Heatmap representations of the average spatial proximity distances between the pairs of all the markers in the multiplexed IMC data are shown.

Supplementary Figs. 15-16). This approach of taking the top nearest neighbor's distances yielded the visualization of spatial proximity between markers in an unbiased manner, whereas calculating the pairwise distance of all cells would be biased by the region area and density of a marker. CD3 and CD4 markers demonstrated high spatial proximity and GrB and CD68 markers for both disease and healthy tonsil datasets (Fig. 6 and Supplementary Figs. 17-19).

**SpatialViz topographic map of tissue layers**. Tissues typically exhibit multiple cell layers that are wrapped around distinct anatomical features. Visualization and quantification of "layering" in the multiplexed data can be informative of tissue architecture that may alter its shape and structure during disease formation. In our tonsil data, around the crypt region of tissues from diseased tonsil specimens, cell-surface antigen marker CD44 exhibited layering formation in both basal, parabasal, and middle layers of the surface and crypt epithelium[39]. While the IHC analysis showed the presence of CD44 as the basal part of the epithelial layers in previous reports, it would have been challenging to visualize the "layering" of the crypt and epithelial regions without the multiplexed protein images. To address this issue, we presented a combination of two-dimensional (2D) spatial topographic and three-dimensional (3D) surface visualization of CD44 positive cells in the form of layered tissue anatomy that was mostly observed in the diseased tonsils but not in normal tonsils (Fig. 7). The topographic plot is classically used for representing geographic layers accurately on a two-dimensional surface using latitudes and longitudes. Akin to this spatial layering graph, visualization of CD44 marker expression levels using a topographic map provided an accurate representation of expression levels as contour plots of 2D images and 3D surface visualization for interactive or animated visualization (Supplementary Movies 1–4).

The 2D intensity plot showed three marker expressions that included CD44 (blue), Pankeratin (red), and GrB (green) (Fig. 7a). In diseased tonsil cases, the crypt region coexisted with the GrB marker. The crypt region border was surrounded by a pankeratin marker that separated the surface epithelia from the crypt region. Around the crypt and epithelial layers, CD44 exhibited a distinct layering near the crypt border, whereas CD44 lacked layering in the normal tonsil. The 2D topography plot indicated the layered CD44 marker distributions in the form of contours (Fig. 7b). The 3D topography plot demonstrated the CD44 marker expression level in a 3D layering graph (Fig. 7c). The average distance between the peak of pankeratin and CD44 markers in the DT2 tonsil was 126-μm, and in the DT3 tonsil was 228-μm. Topographic visuals preserve relative physical distances in 2D/3D representations, and they may experience esthetic variations due to the depth perception.

**SpatialViz reference for multiple tissue organization groups**. One common issue in spatial data analytics is that comparisons across imaging data may be complicated due to the diversity of image heterogeneity. Conventionally, colocalization of markers, the density differences of cells, and differential expression analysis are used to compare multiple spatial maps arising from health and disease conditions using the same marker data. However, a single statistical representation to visualize the differences of spatial complexity in multiplexed datasets would be needed to quantify the differences in tissues from healthy and diseased subjects. To generate a common visual representation for a wide range of spatial tissue architectures, we defined spatial statistical analysis of marker expressions and single-cell positions of the tissue data in a "fixed-node" as part of a spatial reference framework map (Fig. 8a). While a fixed-node landmark network map exists[40], it is limited to analyzing cell population similarity and understanding of cell organization by comparing correlation with known cell phenotypes. For instance, the Scaffold framework provided comparisons of cell structures from different mass cytometry datasets in a graphical representation by analyzing their protein expressions' similarity without any spatial inference. Thus, SpatialViz presented a fixed-node spatial reference framework based on the anatomical characteristic of marker

expressions of multiplexed images to generate reference maps for comparisons across healthy and disease datasets.

The spatial reference maps modeled the distance between cells in pairs of markers and each maker's area ratio from multiplex tissue images. The edges between nodes showed the average distance between markers, while the nodes' size represented the area ratio of a specific marker. We compared the markers' spatial organization in normal and disease tonsil datasets by fixing common markers (DNA1, DNA2, Histone3, CD3, CD4, CD8α, CD20) and let other marker nodes position themselves using force-directed graph algorithm[41]. DNA1, DNA2, and Histone3 nodes have the biggest size across all datasets because of the high area ratio, while GrB and CD68 nodes diameter is larger in disease tonsil than normal tonsil. The edges between CD68, Vimentin, and GrB in disease tonsil also suggest a higher spatial correlation than normal tonsil. These findings correlated with the SpatialViz Anatomy analysis that was shown previously.

Quantification across tissue datasets for marker expression level and cell prevalence area per marker (Fig. 8b) showed consistency in both expression level (circle color) and cell prevalence area (circle area) for healthy and diseased tonsils. Pairwise marker cell distance and a fraction of length <30 μm (Fig. 8c and Supplementary Fig. 20) exhibited higher values for Granzyme B and CD68 in diseased tonsils than healthy tonsils. Clustering of three healthy and three diseased tonsil datasets provided noticeable differences in markers' expression level and markers' cell prevalence area. The mean value of pairwise analysis yielded different cell-to-cell distances and spatial proximity within a fraction of length inferior to 30-μm separation in healthy and disease tonsil data (Supplementary Fig. 21).

## Discussion

SpatialViz uniquely combines the bottom-up and top-down data analyses to decipher the anatomical characteristics such as distinct tissue regions and region-dependent single-cell distributions (Supplementary Fig. 22). The intra-cluster and the inter-cluster analysis revealed the heterogeneity and tissue-scale variance of cellular distributions in tonsil tissues from healthy and diseased subjects. Consistently, the biological finding of GrB and CD86 coexpression was observed in diseased tonsils compared to healthy tonsils. The corresponding spatial proximity maps revealed the relative distances of these two markers with a small average separation measurement and significant spatially resolved shared network maps between the two markers. Diseased tonsils contained large crypt regions on the outer surface epithelium that separates epithelial cells and other immune cell types[39]. The crypt region's presence depended on the image dataset as it was not always present in diseased tonsils (Supplementary Fig. 14). This observation is akin to normal tissues found in tumor biopsies, wherein local regions might have more infection presence than other parts of the tissue. In this SpatialViz analysis, we separated the crypt phenotypes from those without significant crypt formation in the tissue regions. In diseased tonsils that contain the crypt region, the CD44 marker exhibited high expression around the surface epithelium surrounded by pankeratin marker distributions. At the same time, GrB was highly expressed in the crypt region. On the other hand, the CD44 marker high expression area was more randomly spread without obvious layering when crypt regions were not present.

The SpatialViz toolkit proposed in this paper visualized the high-dimensional multiplex imaging data for the anatomical characteristics and spatial relationships among multiple markers. While the presented results were primarily demonstrated to study tonsil biology, these spatial visualization methods can be applied to tumor tissues. The same analysis clusters anatomical regions

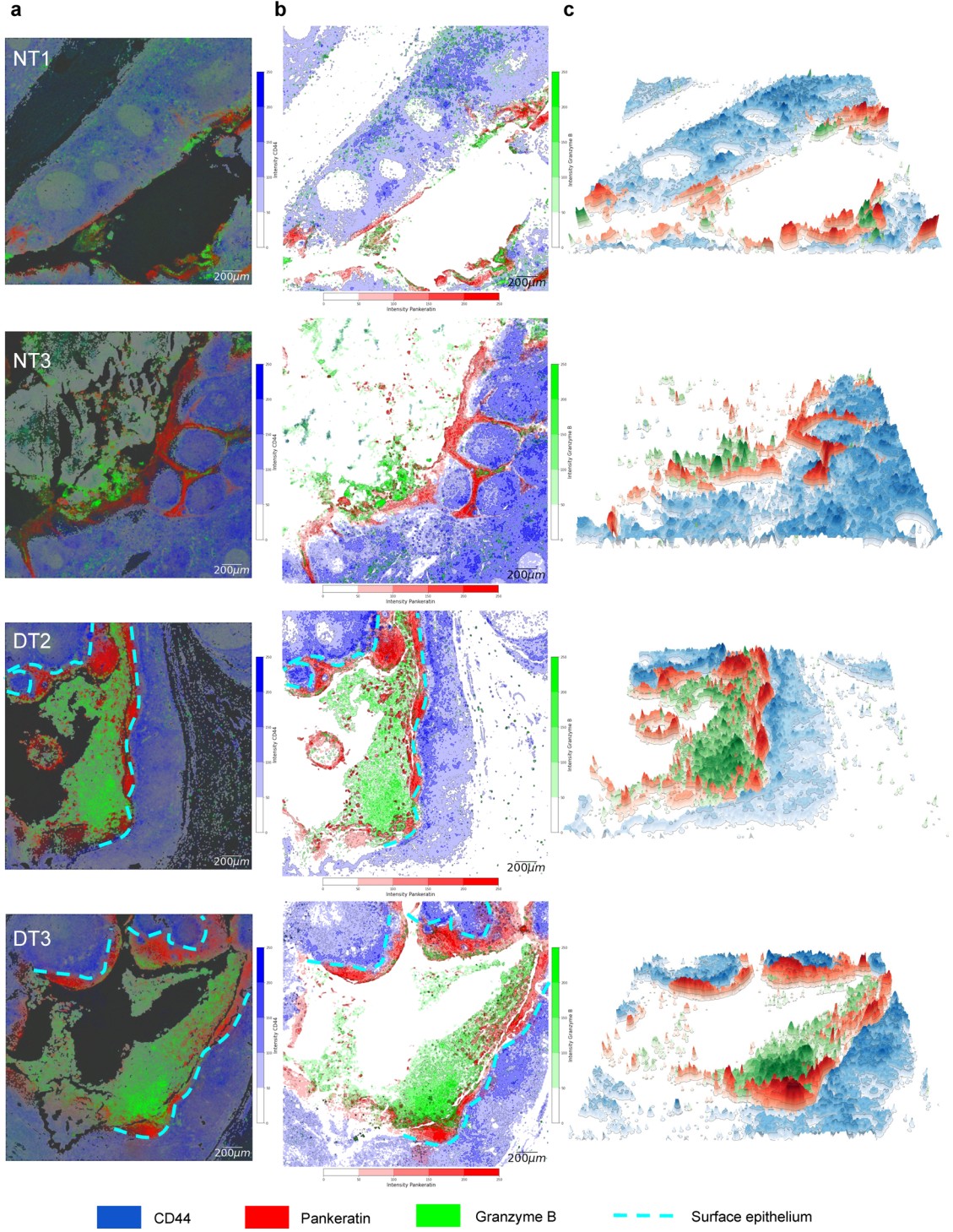

**Fig. 7 Spatial topographic map for two-dimensional (2D) and three-dimensional (3D) visualization of CD44, Pankeratin, and Granzyme B layering in diseased and healthy tonsil tissues. a** 2D intensity plot of overlaid CD44 (blue), Pankeratin (red), Granzyme B (green) markers is shown. The scale bar is 200 μm. **b** 2D topography plot of density maps of markers in the form of contours is presented. The light blue dashed lines denoted the surface epithelial boundary and the crypt region in the disease tonsil. The scale bar is 200 μm. **c** 3D topography plot of the same three markers is shown. Distinct 3D areas with high CD44 and Pankeratin expressions were layered concerning each other's spatial distribution in diseased tonsils, whereas CD44 enrichment was more randomly distributed in normal tonsils. The scale bar is 200 μm.

based on various multiplexed marker expressions and their dominant spatial enrichments in distinct tissue regions, providing the hierarchy of single-cell to anatomy relationships in cancer biopsies. With the addition of patient information, disease stage, and type, SpatialViz can design more personalized treatments using complementary deep learning-based analysis pipelines.

## Methods

**Tonsil tissue preparation and isotope-conjugated antibody labeling**. Normal tonsil and diseased formalin-fixed paraffin-embedded (FFPE) blocks were purchased from a third-party vendor (Biomax US with tissue IDs 1052993.7 and 1051869.3). The FFPE tissue blocks were then cut into serial sections at a thickness of 5-μm and mounted on Superfrost™ Plus Gold Slides (Catalog number: FT4981GLPLUS, ThermoFisher). Three serial tissue slices from each condition

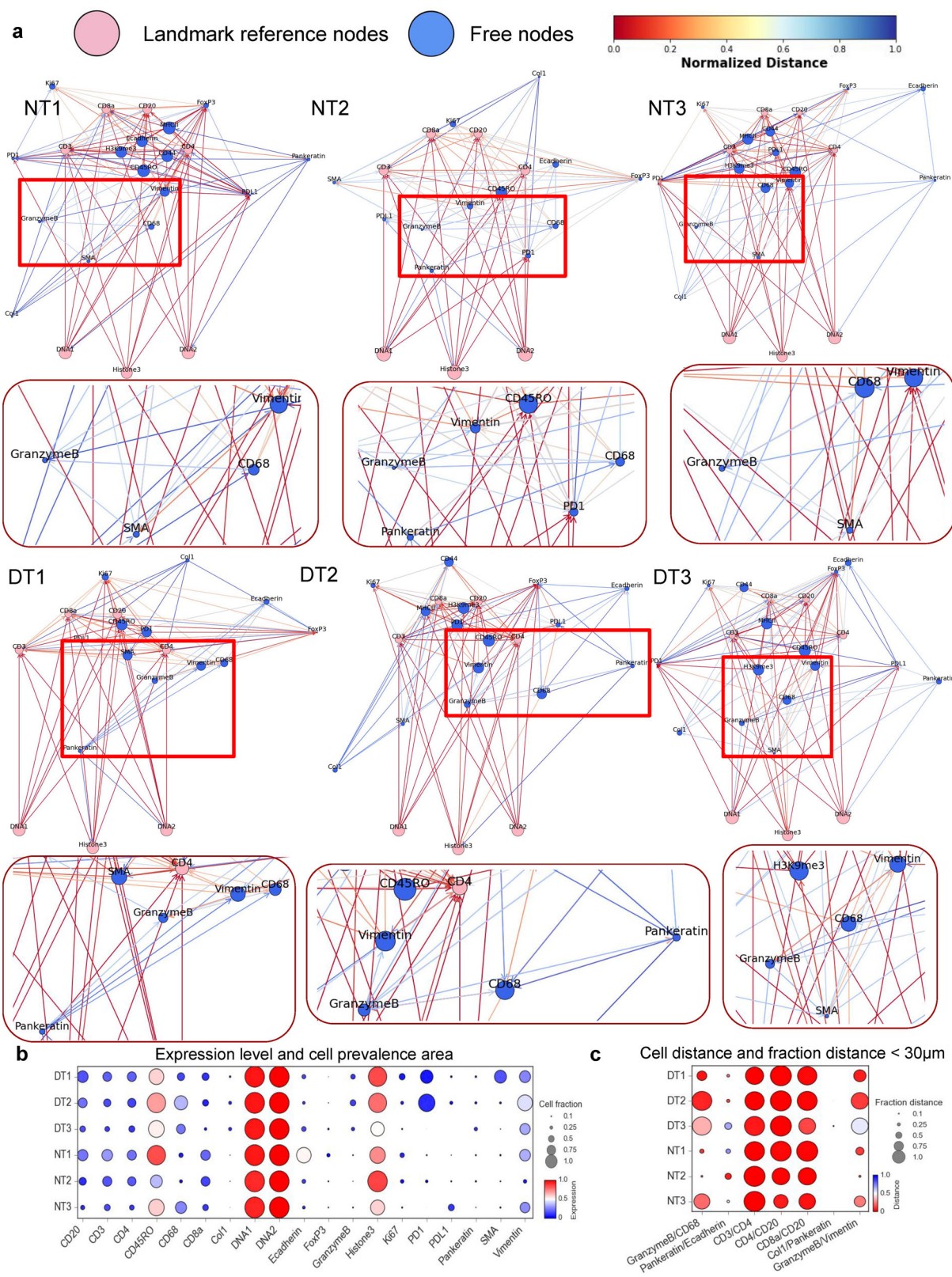

(healthy and diseased) were used for this study. To prepare the sample for labeling and image, they were first deparaffinized by xylene immersion and rehydrated by sequential immersion steps in descending ethanol concentrations (100, 95, 80, 70, and 50%) and a final wash step in deionized water. The tissue slides were then immersed in a basic target retrieval solution with pH = 9 (Catalog number: S2367, Agilent Dako) as per Fluidigm recommendation and left in a pressure cooker on the high-pressure setting (20 min) to achieve the heat-induced epitope retrieval (HIER) process. After the HIER process, the samples were left in the target retrieval

solution for additional 20 min at room temperature. The samples were then dried, and a PAP pen was used to draw a hydrophobic barrier around the specimen on the slide. After this step, a serum-free, ready-to-use protein blocking buffer (Catalog number: X090930-2, Agilent Dako) was applied to the sample for 30 min at room temperature. After three washes of Maxpar PBS (Catalog number: 201058, Fluidigm), the tonsil tissues were then stained with a metal-conjugated antibody cocktail mix for immune markers overnight (4 °C) following Fluidigm labeling protocol (Supplementary Table 1). The tissue slides were then washed with Maxpar

**Fig. 8 Spatial reference framework of multiplied markers in single-cells of pathological clusters in healthy tonsil and chronic tonsillitis cases. a** The spatial reference framework shows the highest five k-nearest neighbor (k-NN) distances for each marker. The pink nodes represent fixed reference landmark markers, whereas blue nodes denote position-free markers, providing a reference of marker alterations from health to disease across tonsil tissue biology. The colors of the edge between the nodes represent the relative nearest neighbor distance among markers for individual cells. **b** Dot plot representations of marker expression levels and cell prevalence areas across three normal datasets and three disease datasets are shown. The circle area represents the cell prevalence area of a specific marker, and the colormap represents the normalized expression level. **c** Dot plot represents the mean value of pairwise marker sets of cell-to-cell distances and fraction of length inferior to 30 μm across three normal tonsil datasets and three diseased tonsil datasets. The circle area represents the fraction of measurement within 30 μm of a specific marker, and the colormap indicates the mean cell-to-cell distance.

PBS (three times), and the nuclear staining was performed using an Iridium-conjugated intercalator (Catalog number 201192A, Fluidigm) for 30 min at room temperature. Finally, slides were washed with three changes of Maxpar PBS and left to dry overnight.

**Imaging and analysis of tonsil tissues**. Three sections from the normal tonsil FFPE block and three additional sections from the diseased tonsil FFPE block were used for this experiment. These samples were first imaged using the bright field imaging setting on the Keyence microscope (BZX 810) to mark the ROIs that were used to reference the IMC signal acquisition. Two different ROIs were chosen from each section. Thus, we had six different ROIs from each condition with the size of 2500 μm × 2500 μm, adding up to a total of 12 ROIs from both healthy and diseased tonsils. Fluidigm's Hyperion imaging system was used to retrieve signals from 18 mass channels associated with biomarkers of interest in addition to two nuclear channels (Supplementary Tables 2-3). After the Hyperion system is done with imaging the chosen areas, it automatically saves the data corresponding to each ROI as a separate MathCad file. These files were first analyzed on the MCD Viewer software (v1.0.560.6) and exported as OME-TIFF 16-bit file format (Supplementary data 1). Each ROI would have 20 different OME-TIFF files, each corresponding to a different mass-channel and its conjugated protein. After a series of optimizations, cellular segmentation masks and single-cell protein expression data were generated using the Cellprofiler (4.0.7) data analysis pipeline as recommended by Fluidigm. All ROIs with their corresponding OME-TIFF files were first imported into CellProfiler. The Metadata function was used to divide the images based on their ROI number, isotope name, and sample name. Then, the NamesAndTypes function was used to match the isotopes' names to their conjugated antibodies and proteins. The "groups" function was used to group individual OME-TIFF images based on their corresponding ROI. Finally, the data analysis pipeline was applied to each ROI file separately.

Data analysis pipeline was generated by adding different CellProfiler modules. In the image processing library, the "ImageMath" module was added and applied to DNA signals from $^{191}$Ir and $^{193}$Ir. This function multiplied the nuclear signal by ten to make the process of nucleus segmentation more efficient. The nucleus signal was then segmented and added using the "IdentifyPrimaryObjects" module in the object processing library pipeline. After a few iterations, the nuclei diameter range was set to be 5–20 pixels. Cell membrane boundaries were segmented using the "IdentifySecondaryObjects" module by expanding the primary objects' pixel size by three. Single-cell protein expression data were extracted from the "MeasureObjectIntensity" module in the measurement library. This data was exported to a spreadsheet in the file format.csv by the module "ExportToSpreadsheet" from the data tools library. Finally, cell masks were generated for later downstream processing using the "ConvertObjectsToImage" module from the object processing library such that each ROI had a separate cell segmentation mask. Finally, ROIs in the form of OME-TIFF data were extracted from MCD Viewer alongside their cell mask generated by CellProlifer were all imported to HistoCAT to analyze correlation and cellular compositions of the tonsil tissue in health and disease state at the subcellular level.

**Pixel-level image clustering using K-means**. Marker images were clustered in an unsupervised manner using the K-means algorithm on each IMC image's grayscale pixel level. K-means clustering was performed using the Scikit-Learn package cluster K-Means in Python with default parameters. From each marker image (2500 × 2500 pixels), we extracted the expression binary mask representing the anatomical region. The binary mask is defined as binary thresholding with a threshold of 60 that was determined experimentally. Each marker mask was then flattened to a single vector (matrix size: 6375000) and stacked together. The resulting matrix (22 × 6375000) is used for K-means clustering using the Scikit-Learn package in Python with default parameters (n_initial = 10, maximum_iterations = 300, tolerance = 1e−4). The K values were chosen from empirical results, given the better separation of images.

**Combined cluster representation plot**. Each cluster marker image was processed using the anatomical clusters' mean values to visualize that marker. Each cluster image is scaled to the 2nd and 98th percentile intensity value and assigned with a unique color using the "gist_rainbow" colormap from the Matplotlib package in

Python. Finally, all the cluster marker images are combined into a single visualization plot to show the anatomical regions of the ROI.

**k-NN distance**. k-NN of distance for each cell was computed using the K-Neighbors Classifier module of the Scikit-Learn package in Python with default parameters. The k-NN classifier was performed using the Scikit-Learn library in Python with default parameters (leaf_size = 30, $p = 2$, metric = 'minkowski', weights = 'uniform'). For each individual single-cell, only ten nearest neighbors were chosen for calculating the pairwise distance between markers.

**Network graph**. Custom-developed Python scripts were used for generating both intra-cluster/inter-cluster spatial network maps and spatial reference maps using the Python NetworkX library. NetworkX is a Python package for exploration and analysis of networks and networks algorithm that provides data structures representing many types of networks, both directed and undirected. Using NetworkX generates various graph formats with flexibility in Python language and connects to other Python packages such as SciPy, NumPy, or Sklearn. From the average of the calculated k-NN distance, the spatial proximity network graph was laid out using the Networkx package[42] in Python with spring layout (k = 0.3 and iteration = 30). The area ratio of the marker determined the size of the nodes. The nodes' color corresponded to the cluster to which they belong, and the weight between the nodes showed the average k-NN distance between two markers or clusters. Edges between the nodes showed the average k-NN distance between two markers or clusters.

**Topographic map**. The 2D intensity map and 2D topographic plot were implemented using the Matplotlib package[43] in Python. For the 2D intensity map, the background was the cell segmentation mask, and original marker-specific expression images were overlaid with the original cell mask. Based on the intensity level, the 3D topography plot was generated using the Plotly package in Python. The height measures in the 3D topography plot represented the pixel intensity distributions of the markers.

**Spatial proximity map**. The spatial proximity maps were composed of two selected components that included visual distance plots of marker pairs and the distances' histogram distribution. Each cell centroid from the first marker (in magenta) was connected to the nearest distance cell in the other marker cell centroids (in green) with a blue line. The cell centroid was obtained using the Measure module of the scikit-image package in Python from the cell segmentation masks. The origin marker cells were represented in magenta, and the destination cells were shown in green. The histogram plot yielded the relative distances between the markers that are expressed on each cell.

**Spatial reference landmarks**. Spatial reference landmarks network was laid out using the Networkx package in Python with spring layout (k = 3.0 and iteration = 5). The landmark nodes in red were fixed positions, and the blue nodes were positioned by both the node size that represented the area ratio and the edge weights calculated by the k-NN distance. For each marker, the average k-NN distances between its cell and other marker cells were computed, and only the highest k-NN that covered a five-cell radius were preserved. The starting of the edges between two nodes indicated the origin marker from which cell distances. The arrow's end corresponded to the markers with the closest k-NN distance from the presumed marker to be the origin. The color of the edges denoted the relative distance between the two markers.

**Statistics and reproducibility**. Images, exhibited in Figs. 1–8 and Supplementary Figs. 1-22, denote data from at least three independent experiments for health and disease. Cellprofiler (Version 4.0.7) was used for processing. Python (Version 3.8.6) processing was performed in the Anaconda environment (Version 4.9.2) using Jupyter notebook. Box plots showed median first and third quartile, minimum and maximum (excluding outliers). Statistical analyses were performed with the Python Scipy package (Version 1.5.2) and HistoCAT (Version 1.76). The two-sample Kolmogorov–Smirnov test was used for data distribution comparison. A $p$-value of <0.001 (***) was considered statistically significant.

**Reporting summary**. Further information on research design is available in the Nature Research Reporting Summary linked to this article.

## Data availability

Supplementary data 1 contains the source data for Fig. 2g-h and Fig. 3a-b. All data and analysis results are available at https://github.com/coskunlab/SpatialViz and https://doi.org/10.5281/zenodo.4662854[44].

## Code availability

The IMC image processing codes are available[44] at https://github.com/coskunlab/SpatialViz and https://doi.org/10.5281/zenodo.4662854.

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

## Acknowledgements

A.F.C. holds a Career Award at the Scientific Interface from Burroughs Wellcome Fund, National Institute of Health K25 Career Development Award (K25AI140783), and a Bernie-Marcus Early-Career Professorship. A.F.C. was supported by start-up funds from the Georgia Institute of Technology and Emory University. This work was performed in part at the Materials Characterization Facility (MCF) at Georgia Tech. The MCF is jointly supported by the GT Institute for Materials (IMat) and the Institute for Electronics and Nanotechnology (IEN), which is a member of the National Nanotechnology Coordinated Infrastructure supported by the National Science Foundation (Grant ECCS-1542174).

## Author contributions

M.A. and T.H. equally contributed to the experiments, data analysis, and writing of this paper. S.C. contributed to the initial experiments. Data acquisition was performed with the help of K.L. and R.B.H. in the IBB core. A.F.C. supervised the project and wrote the paper.

## Competing interests

The authors declare no competing interests.
