## [Peer Review File · Communications Biology]

Reviewers' comments:

Reviewer #1 (Remarks to the Author):

In this manuscript, the authors performed imaging of the normal and diseased tonsil tissues using imaging mass cytometry and visualized the multi-dimensional mass cytometry data with a set of algorithms ("SpatialViz"). The visualization was used for the characterization of the tonsil tissues in health and disease by locating the single cells and providing the 2D or 3D tissue images.

The manuscript provides interesting work in mass cytometry and its data visualization. However, I have some comments that the authors need to address before this manuscript can be accepted for publication in CommsBio.

By using metal labels for the cells, mass cytometry provides more potential labels and distinctive spectra with barely no overlap as compared with fluorescence cytometry. The high-dimension data obtained with mass cytometry motivated the development of visualization tools such as the t-SNE mentioned in the manuscript. More key references for the development of mass cytometric techniques should be included in the introduction part.

Imaging mass cytometry is destructive as compared with fluorescence imaging. For clinical applications, the tissue samples need to be kept while the UV laser exposure vaporized the tissue in mass cytometry. This way the tissue sample can not be re-observed? The staining of the tissue samples and then imaging with optical microscopy are less complex and widely adopted by pathologists. The motivation of this study should be explained better in the introduction part.

Parameters should be provided for the imaging using mass cytometry. How long does it take to get an image for a given tissue size? What's the power of the laser and excitation time?

Will the laser ablation alter the original position of cells in tissue?

It seems the authors has performed optical microscopic imaging of the tissues. Please includes these images and compare with the mass cytometric imaging. What's the resolution of the imaging mass cytometry? Scale bar is lacking in some figures.

In terms of the single cell measurements, related work has been reported as "Learning Single-Cell Distances from Cytometry Data". Please comment.

Figure 1e: Hard to tell which color is for the individual marker.

For the 3D reconstruction, how to keep the relative position of single cells on different layers?

Reviewer #2 (Remarks to the Author):

The paper submitted by Coskun and colleagues presents a SpatialViz, a new tools for the analysis expression profile at tissue level.

The developed method is important for the analysis of spatial complexity in multiplexed datasets based on quantitative observables. Although the paper provides many quantitative details of the differences found tissues from normal and diseased subjects, at the current stage it is difficult to understand the level of generalization of the method.

Thus, I believe that the paper can be considered for publication after addressing the points reported below.

Major Revisions

1) The Methods section of the manuscript needs to be improved. More details about pipeline and the analysis should be provided.

- More details about the pipeline used for extracting the regions of interest should be included.

More information about the standard data analysis pipeline in Cell Profiler and HistoCAT should be reported.

- Little information is provided about the K-means clustering analysis. What are the parameters used for the optimization of the K-Neighbors Classifier module in scipy?
- Please provide a better description input features used for the classification.
- Why did you use Gephi Force Atlas algorithm for calculating the average k-NN distance? Would not be better to have a unique pipeline with using python libraries?

2) In the method section a description of the dataset for testing the method should be improved.

- How many subjects were screened? Did you use 3 tissue slices for each sample?
- Are the number of subjects large enough for the generalization of the method?
- Could you include a quantitative analysis of the consistency in the results in the different tissue slices?

3) In the GitHub page with the SpatialViz codes and datasets poor information about the content of the files is included. Please improve the description of the files and include the files with the data divided by subjects and slices.

Reviewer #1 (Remarks to the Author):

In this manuscript, the authors performed Imaging of the normal and diseased tonsil tissues using imaging mass cytometry and visualized the multi-dimensional mass cytometry data with a set of algorithms ("SpatialViz"). The visualization was used for the characterization of the tonsil tissues in health and disease by locating the single cells and providing the 2D or 3D tissue images.

The manuscript provides a computational framework in imaging mass cytometry for studying tonsil biology in health and disease. We thank the reviewer for acknowledging our research work's potential, including our novel imaging and data visualization tools.

However, I have some comments that the authors need to address before this manuscript can be accepted for publication in CommsBio. By using metal labels for the cells, mass cytometry provides more potential labels and distinctive spectra with barely no overlap as compared with fluorescence cytometry. The high-dimension data obtained with mass cytometry motivated the development of visualization tools such as the t-SNE mentioned in the manuscript. More key references for the development of mass cytometric techniques should be included in the introduction part.

We agree with the reviewer that the imaging mass cytometry (IMC) technology has advantages over routine immunofluorescence and fluorescence cytometry. We refer to the edited introduction in the main manuscript with additional text detailing the benefit of using IMC to visualize proteins in-situ in archived patients samples.

We have added the following paragraph to the introduction to clarify this issue (lines 72-99):

"Although this multiplexed power may be partially obtained by fluorescence cytometry, the fluorophores used to detect biomarkers of interest have overlapping spectra leading to signal spillover between target and non-target detectors. That necessitates the use of additional signal processing techniques to correct for fluorophores signals in non-target detectors, often referred to as spillover compensation. Signal compensation can be more complicated and often lead to erroneous results.^{15,16} Another shortcoming of cell suspension cytometry techniques is its inability to analyze biomarkers in situ from patients' samples, resulting in the loss of the precious spatial single-cell data. Several fluorescence-based multiplexed techniques were later developed and tested for patients' archived FFPE samples, including cyclic immunofluorescence.¹⁷ These methods rely on antibody staining, bleaching, and imaging cycles that suffer from sample autofluorescence, epitope loss, and laborious procedures. On the other hand, IMC detects the abundance of metal isotopes conjugated to antibodies against multiple biomarkers. These lanthanide isotopes are rare elements that would not be found in biological samples, resulting in high signal specificity.¹⁸ The signal is further amplified by polymeric metal-chelating reagents, allowing a range of proteins to be detected. Preserving in situ cell-cell interaction and tumor microenvironments, IMC

*relates the metal isotope abundance to their pixel location, providing images of multiple proteins simultaneously in archived patients samples at subcellular resolution (1- μm).*¹⁸⁻²⁰

IMC is widely used in many applications, including drug testing and tumor studies. IMC was previously used to investigate the potential benefit of the chemotherapeutic agents on three different cancer types. A multiplex antibody panel was designed to target several proteins associated with common cancer oncoproteins and other markers to detect T cell infiltration (CD8) and the epithelial organization (Beta-catenin and Pan-cytokeratin)²¹. Further, IMC was used to analyze more than 15 proteins associated with different stages of Multiple sclerosis (MS), identifying a subset of T cell phenotypes associated with different MS stages²². Besides, IMC was used to decipher the breast cancer tumors using a panel of 35 biomarkers associated with breast cancer subtypes, grades, signaling, oncogenes, and epigenetics. Different breast cancer subtypes were found to have a strong correlation with survival. Remarkably, tumor stroma is highly infiltrated with yielded low survival rates.¹⁹ IMC revealed mRNA-to-protein associations at the subcellular level in breast cancer patient samples.²³ RNA probes and their encoding protein antibodies were conjugated to different metal isotopes to study their correlation at the population level. This growing body of discoveries highlights the importance of IMC technology to reveal interdependencies among cell types and provide spatial maps for multiple subcellular resolution parameters."

Imaging mass cytometry is destructive as compared with fluorescence imaging. For clinical applications, the tissue samples need to be kept while the UV laser exposure vaporized the tissue in mass cytometry. This way the tissue sample can not be re-observed? The staining of the tissue samples and then Imaging with optical microscopy are less complex and widely adopted by pathologists. The motivation of this study should be explained better in the introduction part.

We agree with the reviewer that IMC relies on the UV laser ablation that vaporizes the tissue to ionize the metal isotopes and analyze their abundance based on their time-of-flight. Since the tissue is physically ablated, re-observing is not possible. However, this problem can be overcome by using serial sections from the patients' biopsy. This approach is akin to clinical practice. Biopsy samples used for immune-histological analysis cannot be re-observed, and pathologists often rely on sequential tissue sections to observe more biomarkers of interest. This process is often laborious and susceptible to the variability of interpretation between observers. Thereby, IMC outperforms the routine IHC as it provides a high multiplex power in one experiment run with high sensitivity and specificity from up to 40 markers in one tissue specimen.

Reference: *Tan, W. C. C. et al. Overview of multiplex immunohistochemistry/immunofluorescence techniques in the era of cancer immunotherapy. Cancer Commun (Lond) 40, 135–153 (2020).*

Parameters should be provided for the Imaging using mass cytometry. How long does it take to get an image for a given tissue size? What's the power of the laser and excitation time?

We thank the reviewer for this suggestion. All changes are reflected in the supplementary notes section.

Details about the Fluidigm Imaging Mass Cytometer

ROIs size and Imaging Time:

Regions of interest (ROIs) chosen from the tonsil samples were approximately 2500 μm x 2500 μm . The laser type is Nd: YAG with a wavelength of 213-nm. The energy output was less than or equal to 3 μJ . It takes 2 hours to image 1 mm^2 ; thereby, it took approximately 12.5 hours to acquire individual ROIs on the tonsil sections.

Description	System Specifications
Laser type	Nd: YAG
Wavelength	213 nm
Energy Output	3 μJ
Tissue thickness for full ablation	$\leq 7\mu\text{m}$ thickness
Sample size	$\geq 15 \text{ mm} \times 45 \text{ mm}$
Scan Area	$\geq 1 \text{ mm}^2 / 2\text{Hr}$ (@200Hz)

Supplementary Table 2. System specification for the Hyperion imaging system by Fluidigm.

Slide	Imaging Time (H: M:S)	Laser Power (dB)	Ablation Power (Hz)
Normal Tonsil 1	9:59:51	2.5	200
Normal Tonsil 2	9:59:51	2.5	200
Normal Tonsil 3	9:59:51	2	200
Diseased Tonsil 1	7:45:34	2	200

Diseased Tonsil 2	9:59:51	2	200
Diseased Tonsil 3	9:59:51	2.5	200

Supplementary Table 3. Experimental details regarding samples, imaging time, laser power, and ablation power were used to obtain the paper's data.

Since these product specifications can vary between different experimental runs, we also included some of the parameters associated with two experimental runs for one normal tonsil sample and one diseased tonsil sample. As shown in the snapshots below, different ablation energy values were tried out before choosing the ROIs' final values. This optimization process was done by testing out different values of ablation energies on smaller regions 100 μm x 100 μm instead of 2500 μm x 2500 μm . The value that resulted in a strong signal with the minimum background was finally chosen for the bigger ROIs.

Will the laser ablation alter the original position of cells in tissue?

We thank the reviewer for this insightful comment. IMC technique was validated against immunofluorescence to confirm that cells remain stationary during the imaging, maintaining cell morphology and cell-cell interaction. Bodenmiller's lab is a worldwide pioneer to adopt IMC, which he detailed in his invited talk (host is Dr. Coskun, senior author of this paper) found at the link below. Dr. Bodenmiller showed an IF image and an IMC image from the same cancer tissue (**Figure 1**) to show that this novel technique can capture native tissue architecture without altering cells' position or cytoskeletal proteins' organization.

Imaging mass cytometry images are similar
to fluorescence microscopy images

Figure 1. IMC and Optical imaging of immunolabeled breast cancer tissue section.

Data is from the [Minutes 14.25 -Multiscale analysis of in situ tumor biology towards precision medicine](https://www.youtube.com/watch?v=IU_RmHIMz_I&%3Bt=65s)
(https://www.youtube.com/watch?v=IU_RmHIMz_I&%3Bt=65s)

It seems the authors has performed optical microscopic Imaging of the tissues. Please includes these images and compare with the mass cytometric Imaging. What's the resolution of the imaging mass cytometry? Scale bar is lacking in some figures.

We thank the reviewer for this helpful comment. We imaged an area of 2.5 mm x 2.5 mm in tonsil tissues using 2,500 x 2,500 pixels, yielding 1-µm per pixel. In the meantime, the IMC system (Technical spec, page7) focuses the laser beam to 1-µm spot size, limiting the ultimate resolution of multiplexed imaging. Pixel sampling analysis and laser focal area determine the imaging mass cytometry system's physical resolution to be 1-µm. As noted, the comparisons of optical images and IMC were previously reported in the literature, agreeing well with each other. Thus, IMC shows a performance of a 10× optical microscope in the range of micron spatial details. Further optical characterization of imaging optics is beyond the scope of this current work that specifically reports multiplexed data and SpatialViz data visualization methods.

Besides, all figures in the main manuscript and supplementary have been updated with their appropriate scale bars.

In terms of the single cell measurements, related work has been reported as "Learning Single-Cell Distances from Cytometry Data". Please comment.

We appreciate the reviewer's insight into this interesting field of research. "Learning Single-Cell Distances from Cytometry Data" presents a learning method using the Mahalanobis distance metric derived from the data. From labeled single-cell cytometry data, this learning metric can be used for the classification of single-cell differences. Here, the Mahalanobis distance metric better separates cell populations compared with the Euclidean distance metric. There are similarities in capturing cell types from various biomarkers. While the previous cytometry data was performed in cell suspensions without spatial cell positions, and our approach covers imaging cytometry and data classification in physical spatial distances. Specifically, we have used k-nearest neighborhoods (KNN) to determine the spatial distances between neighboring using spatial proximity analysis of marker pairs in the multiplexed data.

Figure 1e: Hard to tell which color is for the individual marker.

We thank the reviewer for the detailed feedback. Figure 1e has been updated with individual tSNE plots for eight markers in the normal and diseased condition. These markers were also highlighted in the Phenographs for each sample, along with their corresponding cellular phenotypes.

For the 3D reconstruction, how to keep the relative position of single cells on different layers?

We appreciate the reviewer's question about explaining the 3D reconstruction cell position. From 2D marker images (defined on the x and y-axis), we reconstruct the 3D marker images by keeping each image's x and y

coordinates and adding z-axis coordinates representing the same intensity level. The z coordinates define a 3D surface plot that shows the variation of the expression level. This process is shown in the figure below.

To visualize the bordering of the markers, we overlay markers on the same x and y coordinates. The 3D visualization lets us better distinguish the bordering effect of the markers. When changing the camera's position in the 3D coordinates, each cell's relative position is unaltered. Still, the perceived depth is modified as the original point of view angle is changed, as illustrated below.

We present the absolute distance between two cells with different camera positions in the 3D coordinates. In the figure below, l_1 and l_2 have the same exact length (20- μm). But depending on the angle of the point of view, the depth perception is changed. In the left figure, the angle of point of view is equal to 0 degrees (directly perpendicular from the center of the image), and this is equivalent to the usual 2D image. On the other hand, the right figure has a positive angle. The perception is changed, enlarging the relative distance at the bottom of the image and decreasing the relative distance in the second image's top part.

We have added the following to the manuscript to clarify this issue:

Topographic visuals preserve relative physical distances in 2D/3D representations, and they may experience aesthetic variations due to the depth perception.

Reviewer #2 (Remarks to the Author):

The paper submitted by Coskun and colleagues presents a SpatialViz, a new tools for the analysis expression profile at tissue level.

The developed method is important for the analysis of spatial complexity in multiplexed datasets based on quantitative osservables. Although the paper provides many quantitative details of the differences found tissues from normal and diseased subjects, at the current stage it is difficult to understand the level of generalization of the method.

Thus, I believe that the paper can be considered for publication after addressing the points reported below.

Major Revisions

1) The Methods section of the manuscript needs to be improved. More details about pipeline and the analysis should be provided.

- More details about the pipeline used for extracting the regions of interest should be included. More information about the standard data analysis pipeline in Cell Profiler and HistoCAT should be reported.

We thank the editor for the time and effort. All suggestions have been addressed in the material and methods section. The paragraph below is included in the methods section of the revised manuscript in response to this comment (lines 416-428).

Data analysis pipeline was generated by adding different CellProfiler modules. In the image processing library, the "ImageMath" module was added and applied to DNA signals from Ir191 and Ir193. This function multiplied the nuclear signal by ten to make the process of nucleus segmentation more efficient. The nucleus signal was then segmented and added using the "IdentifyPrimaryObjects" module in the object processing library pipeline. After a few iterations, the nuclei diameter range was set to be 5-20 pixels. Cell membrane boundaries were segmented using the "IdentifySecondaryObjects" module by expanding the primary objects' pixel size by three. Single-cell protein expression data were extracted from the "MeasureObjectIntensity" module in the measurement library. This data was exported to a spreadsheet in the file format .csv by the module "ExportToSpreadsheet" from the data tools library. Finally, cell masks were generated for later downstream processing using the "ConvertObjectsToImage" module from the object processing library such that each ROI had a separate cell segmentation mask. Finally, ROIs in the form of OME-TIFF data were extracted from MCD viewer alongside their cell mask generated by CellProlifer were all imported to HistoCAT to analyze correlation and cellular compositions of the tonsil tissue in health and disease state at the subcellular level.

- Little information is provided about the K-means clustering analysis. What are the parameters used for the optimization of the K-Neighbors Classifier module in scipy?

These edits have been inserted in the main manuscript in the materials and methods section (lines 429-447).

Pixel-level image clustering using K-Means: *Marker images were clustered in an unsupervised manner using the K-means algorithm on each IMC image's grayscale pixel level. K-means clustering was performed using the Scikit-Learn package cluster K-Means in Python with default parameters. From each marker image (2,500x2,500 pixels), we extracted the expression binary mask representing the anatomical region. The binary mask is defined as binary thresholding with a threshold of 60 that was determined experimentally. Each marker mask was then flattened to a single vector (matrix size: 6375000) and stacked together. The resulting matrix (22 × 6375000) is*

used for K-means clustering using the Scikit-Learn package in Python with default parameters ($n_initial = 10$, $maximum_iterations = 300$, $tolerance = 1e-4$). The K values were chosen from empirical results, given the better separation of images.

k-NN distance: *K-nearest neighbors (k-NN) of distance for each cell was computed using the K-Neighbors Classifier module of the Scikit-Learn package in Python with default parameters. The KNN classifier was performed using the Scikit-Learn package in Python with default parameters ($leaf_size = 30$, $p=2$, $metric='minkowski'$, $weights='uniform'$). For each individual single-cell, only ten nearest neighbors were chosen for calculating the pairwise distance between markers.*

- Please provide a better description input features used for the classification.

The same as above was included in the methods section to describe the classification approach (lines 429-437).

Pixel-level image clustering using K-Means: *Marker images were clustered in an unsupervised manner using the K-means algorithm on each IMC image's grayscale pixel level. K-means clustering was performed using the Scikit-Learn package cluster K-Means in Python with default parameters. From each marker image (2,500x2,500 pixels), we extracted the expression binary mask representing the anatomical region. The binary mask is defined as binary thresholding with a threshold of 60 that was determined experimentally. Each marker mask was then flattened to a single vector (matrix size: 6375000) and stacked together. The resulting matrix (22×6375000) is used for K-means clustering using the Scikit-Learn package in Python with default parameters ($n_initial = 10$, $maximum_iterations = 300$, $tolerance = 1e-4$). The K values were chosen from empirical results, given the better separation of images.*

- Why did you used Gephi Force Atlas algorithm for calculating the average k-NN distance? Would not be better to have a unique pipeline with using python libraries?

We appreciate the reviewer's insight, and we agree that using a unique pipeline using python libraries is more suited for the visualizations. Therefore, instead of using the Gephi Force Atlas algorithm, custom-developed python scripts were used in the revised manuscript and associated Figures. The following paragraph is now included in the revised manuscript's methods section (lines 448-457).

Network graph: *Custom-developed Python scripts were used for generating both intra-cluster/inter-cluster spatial network maps and spatial reference maps using the Python NetworkX library. NetworkX is a Python package for exploration and analysis of networks and networks algorithm that provides data structures representing many types of networks, both directed and undirected. Using NetworkX generates various graph formats with flexibility in Python language and connects to other Python packages such as SciPy, NumPy, or Sklearn. From the average of the calculated k-NN distance, the spatial proximity network graph was laid out*

using the Networkx package ⁴² in Python with spring layout ($k=0.3$ and $\text{iteration}=30$). The area ratio of the marker determined the size of the nodes. The nodes' color corresponded to the cluster to which they belong, and the weight between the nodes showed the average k -NN distance between two markers or clusters. Edges between the nodes showed the average k -NN distance between two markers or clusters.

2) In the method section a description of the dataset for testing the method should be improved.

- How many subjects were screened? Did you used 3 tissue slices for each sample?

We thank the editor for the time and effort. All suggestions have been addressed in the material and methods section. The paragraph below is included in the revised manuscript's materials and methods section (lines 399-415).

"Three sections from the normal tonsil FFPE block and three additional sections from the diseased tonsil FFPE block were used for this experiment. These samples were first imaged using the bright field imaging setting on the Keyence microscope (BZX 810) to mark the regions of interest (ROIs) that will later be used to reference the IMC signal acquisition. Two different ROIs were chosen from each section. Thus, we had six different ROIs from each condition with the size of 2500 um x 2500 um, adding up to a total of 12 ROIs from both healthy and diseased tonsils. Fluidigm's Hyperion imaging system was used to retrieve signals from 18 mass-channels associated with biomarkers of interest in addition to 2 nuclear channels. After the Hyperion system is done with imaging the chosen areas, it automatically saves the data corresponding to each ROI as a separate MathCad file. These files were first viewed on the MCD viewer software (v1.0.560.6) and exported as OME-TIFF 16-bit file format such that each ROI would have 20 different OME-TIFF files, each corresponding to a different mass-channel and its conjugated protein. After a series of optimizations, cellular segmentation masks and single-cell protein expression data were generated using the Cellprofiler (4.0.7) data analysis pipeline as recommended by Fluidigm. All ROIs with their corresponding OME-TIFF files were first imported into CellProfiler. The Metadata function was used to divide the images based on their ROI number, isotope name, and sample name. Then, the NamesAndTypes function was used to match the isotopes' names to their conjugated antibodies and proteins. The "groups" function was used to group individual OME-TIFF images based on their corresponding ROI. Finally, the data analysis pipeline was applied to each ROI file separately."

- Are the number of subjects large enough for the generalization of the method?

We appreciate the reviewer's question about the generalizability of the method. Here, we show the visualization across tissue samples of Lung cancer for different stages. The H&E image is shown for four cancer stages.

Then we apply our anatomical clustering pipeline to color and visualize different tissue regions in thirteen lung cancer biopsies in the microarray format. SpatialViz results are similar to those obtained in the tonsil sections used in this paper. We will not include this data in our revised manuscript because it is beyond the scope of this presented work related to tonsil tissue biology.

- Could you include a quantitative analysis of the consistency in the results in the different tissue slices?

We show the following dot plot representation of markers expression level and cell prevalence area for all 12 ROIs to analyze the different tissues' results. The circle area represents the cell prevalence area of a specific marker, and the colormap represents the normalized expression level.

The same process is generated without the DNA1, DNA2, and Histone3 markers to distinguish other markers in the figure below.

To look at consistency across the dataset, we performed hierarchical clustering of the ROI from the markers area ratio and expression level.

We also performed hierarchical clustering of the ROI from dot plot representation of spatial proximity analysis using pairwise marker mean cell-to-cell distance and fraction of distance inferior to 30- μ m.

We have added the following to the manuscript (lines 327 -334).

Quantification across tissue datasets for marker expression level and cell prevalence area per marker (Fig. 8b) showed consistency in both expression level (circle color) and cell prevalence area (circle area) for healthy and diseased tonsils. Pairwise marker cell distance and a fraction of length less than 30- μ m (Fig. 8c and Supplementary Fig. 20) exhibited higher values for Granzyme B and CD68 in diseased tonsils than healthy tonsils. Clustering of three healthy and three diseased tonsil datasets provided noticeable differences in markers' expression level and markers' cell prevalence area. The mean value of pairwise analysis yielded different cell-to-cell distances and spatial proximity within a fraction of length inferior to 30- μ m separation in healthy and disease tonsil data (Supplementary Fig. 21).

3) In the GitHub page with the SpatialViz codes and datasets poor information about the content of the files is included. Please improve the description of the files and include the files with the data divided by subjects and slices.

We appreciate the reviewer's suggestions about the documentation on the GitHub page with the SpatialViz codes. The GitHub page (<https://github.com/coskunlab/SpatialViz>) has now been updated with a detailed README file showing the codes' organization, example images, and step by step usage guidelines.

REVIEWERS' COMMENTS:

Reviewer #1 (Remarks to the Author):

The authors have replied to all my previous comments and made changes in the manuscript. I recommend the manuscript in its current format to be accepted.

Reviewer #2 (Remarks to the Author):

In the final version of the manuscript the authors addressed the points raised in my previous review.

Thus I suggest the editor to accept the paper for publication.